# CsrA Regulates Swarming Motility and Carbohydrate and Amino Acid Metabolism in *Vibrio alginolyticus*

**DOI:** 10.3390/microorganisms9112383

**Published:** 2021-11-18

**Authors:** Bing Liu, Qian Gao, Xin Zhang, Huizhen Chen, Ying Zhang, Yuehong Sun, Shan Yang, Chang Chen

**Affiliations:** 1CAS Key Laboratory of Tropical Marine Bio-Resources and Ecology (LMB), Guangdong Provincial Key Laboratory of Applied Marine Biology (LAMB), South China Sea Institute of Oceanology, Chinese Academy of Sciences, Guangzhou 510301, China; liubing19@mails.ucas.ac.cn (B.L.); gaoqian21@mails.ucas.ac.cn (Q.G.); zhangxin174@mails.ucas.ac.cn (X.Z.); chenhuizhen20@mails.ucas.ac.cn (H.C.); zhangy@scsio.ac.cn (Y.Z.); 2College of Earth and Planetary Sciences, University of Chinese Academy of Sciences, Beijing 101408, China; 3School of Environment, South China Normal University, Guangzhou 510006, China; yuehong.sun@m.scnu.edu.cn; 4College of Marine Sciences, South China Agricultural University, Guangzhou 510600, China; ys1909scau@163.com; 5Xisha and Nansha Ocean Observation and Research Station, South China Sea Institute of Oceanology, Chinese Academy of Sciences, Guangzhou 510301, China

**Keywords:** CsrA, *Vibrio alginolyticus*, motility, carbohydrate and amino acid metabolism, mRNA stability

## Abstract

*Vibrio alginolyticus*, like other *vibrio* species, is a widely distributed marine bacterium that is able to outcompete other species in variable niches where diverse organic matters are supplied. However, it remains unclear how these cells sense and adjust metabolic flux in response to the changing environment. CsrA is a conserved RNA-binding protein that modulates critical cellular processes such as growth ability, central metabolism, virulence, and the stress response in *gamma*-proteobacteria. Here, we first characterize the *csrA* homolog in *V. alginolyticus*. The results show that CsrA activates swarming but not swimming motility, possibly by enhancing the expression of lateral flagellar associated genes. It is also revealed that CsrA modulates the carbon and nitrogen metabolism of *V. alginolyticus*, as evidenced by a change in the growth kinetics of various carbon and nitrogen sources when CsrA is altered. Quantitative RT-PCR shows that the transcripts of the genes encoding key enzymes involved in the TCA cycle and amino acid metabolism change significantly, which is probably due to the variation in mRNA stability given by CsrA binding. This may suggest that CsrA plays an important role in sensing and responding to environmental changes.

## 1. Introduction

*Vibrio alginolyticus* is a common gram-negative opportunistic pathogen present in a large variety of marine animals and human beings. It is widely distributed in marine waters, sediments, and on the surfaces of marine organisms. It is one of the predominant cultivable bacteria in the tropical or subtropical coastal waters, and outbreaks of this species often cause great losses in the marine aquaculture industry [1,2,3,4,5,6,7,8,9]. A distinguishing feature of *V. alginolyticus* is its ability to grow rapidly in the environment where abundant organic matters are supplied, which provides it with advantages over other bacteria. However, it remains unclear how it incorporates metabolic pathways in response to diverse carbon and nitrogen sources.

The carbon storage regulator (Csr) or repressor of stationary phase metabolites (Rsm) system of *gamma*-proteobacteria is among the most complex and best-studied post-transcriptional regulatory systems [10]. CsrA is an RNA binding protein that acts by binding to a consensus sequence containing the GGA motif of single-stranded RNA, thus affecting the translation and/or stability of a wide range of target mRNAs. Therefore, it plays an important role in regulating metabolism, virulence, motility, and other cellular processes [11,12,13,14,15,16]. CsrA activity is controlled by multiple small noncoding RNAs (sRNAs), namely CsrB/CsrC in *Escherichia coli* and CsrB/CsrC/CsrD in *V. cholerae*, which contain dozens of high affinity GGA motifs that compete with those of mRNA targets [17,18,19,20]. 

Only a few studies concerning CsrA have been carried out in *Vibrio*, among which *V. cholerae* has been most extensively explored. In *V. cholerae*, CsrA is indispensable for cell survival and is critical for quorum sensing, virulence, and amino acid metabolism [21,22]. RNA-seq analysis showed that CsrA affects the transcripts of as many as 22% of the genes in the *V. cholera* genome, whose functions include amino acid transport and metabolism, central metabolism, lipid metabolism, iron uptake, and flagellum-dependent motility. [23]. 

In the present study, we constructed a point-mutation strain and an overexpression strain of *csrA* and subsequently compared their growth ability, motility, and metabolism of various carbon and nitrogen sources. We showed that CsrA activates swarming rather than swimming motility, possibly by enhancing the expression of lateral flagellar associated genes. Additionally, CsrA is extensively involved in carbohydrate and amino acid metabolism, as evidenced by changes in the growth kinetics of different carbon and nitrogen sources when *csrA* is altered or overexpressed. In addition, the transcript abundance and mRNA stability of related genes were quantified. The transcripts of the genes encoding key enzymes involved in the pyruvate and TCA cycles and amino acid metabolism were found to be significantly changed, probably due to the variation in mRNA stability given by CsrA binding. These results suggest that CsrA is a global regulator that modulates motility and central metabolism in *V. alginolyticus*, and this study may shed light on the regulatory role of CsrA in *V. alginolyticus*.

## 2. Materials and Methods

### 2.1. Bacterial Strains, Plasmids, and Media

All bacterial strains and plasmids used in this study are listed in Table 1. All strains were maintained at −80 °C in Tryptic Soy Broth (TSB) (BD, USA) plus 25% glycerol. *V. alginolyticus* and derivatives were routinely cultured in TSB or Luria-Bertani (LB) broth (1% tryptone, 0.5% yeast extract, 1% NaCl [wt/vol]) (VWR International, LLC, Poland) plus 2.5% NaCl at 30 °C. *Escherichia coli* strains were cultured in LB medium supplemented with appropriate antibiotics at 37 °C. For the selection of transconjugants, TCBS medium (HuanKai, Guangzhou, China) was used with 5 μg/mL chloramphenicol (Cm) and 0.2% D-glucose. The amplified fragments were cloned into a suicide vector, pSW7848 [24] encoding the *ccdB* toxin gene under the control of an arabinose-inducible and glucose-repressible promoter. To select transconjugants that had undergone plasmid excision and allelic exchange, TCBS medium plus 0.2% arabinose plus 5 μg·mL^−1^ chloramphenicol (Cm) or TCBS medium plus 0.2% arabinose alone was used to induce the *ccdB* gene and to select bacteria that had lost the inserted plasmid. Unless otherwise indicated, M63 minimum medium [25] was made according to the following recipe: 3 g·L^−1^ KH_2_PO_4_, 7 g·L^−1^ K_2_HPO_4_, 2 g·L^−1^ (NH_4_)_2_SO_4_, 0.5 × 10^−3^ g·L^−1^ FeSO_4_, 30 g·L^−1^ NaCl, 2 × 10^−3^ M MgSO_4_, 5 × 10^−3^ g·L^−1^ thiamine, and 4 g·L^−1^ D-glucose. Antibiotics were used at the following concentrations: chloramphenicol (Cm) at 5 μg/mL for *V. alginolyticus* and 20 μg/mL for *E. coli*. When necessary, diaminopimelate (DAP) was added to the growth media at a final concentration of 0.3 mM.

### 2.2. Mutant and Plasmid Construction

To create the *V. alginolyticus csrA* point mutation, two 1000-bp PCR products flanking the target of interest were generated by splice overlap PCR using two pairs of the primers *csrA*R6H-up_fwd and *csrA*R6H-up_rev, *csrA*R6H-down_fwd and *csrA*R6H-down_rev respectively (Appendix A). These two flanking products were engineered to have overlapping sequences, enabling the products to anneal and generate a ligated product. The amplified linearized plasmid pSW7848 (which can only replicate in a Pir-positive (Pir^+^) strain [24]) and purified PCR products corresponding to 1000 bp upstream and downstream of the target gene were mixed using the MultiS One Step Cloning Kit (Vazyme Biotech Co., Ltd., Nanjing, China), generating plasmid pSW7848-*csrA*.R6H (Table 1), which was then confirmed using the primers annotated Del-check-pSW7848-F and Del-check-pSW7848-R.

To generate deletions, pSW7848-*csrA*.R6H with *E. coli* GEB883 as an intermediate host contained the upstream and downstream flanking sequences of the gene of interest; the gene itself was not included. The recombinant plasmid was transferred by conjugation from strain GEB883 (Table 1) to *V. alginolyticus* wild-type strain ZJ-T before allelic exchange. Conjugations and selection of mutants were carried out as follows. The donor strain (GEB883 with the mobilizable plasmid) and the recipient strain *V. alginolyticus* ZJ_T were grown to the stationary phase in LB plus DAP plus Cm (*E. coli*) and in LBS (ZJ_T) at 37 and 30 °C, respectively. Overnight cultures of the donor and recipient strains were diluted 100-fold in broth and grown to an optical density of 0.3 at 600 nm. A 1.5 mL volume of donor cells was centrifuged and washed twice with LB and then resuspended in 100 μL LBS, whereas recipient cells were centrifuged and resuspended in 250 μL of LBS. For conjugation, 10 μL of donor cells and 50 μL of concentrated recipient cells were spotted on an LBS agar plate plus 0.3 mM DAP overnight at 30 °C.

Cells were then resuspended from the agar plate in 1 mL of LBS, centrifuged, resuspended in 100 μL of LBS medium, and spread for selection of the exconjugants on a TCBS-plus-Cm agar plate in the presence of 0.2% glucose before incubation at 30 °C until colonies appeared. Potential exconjugants were purified twice on the same selective medium at 30 °C and then restreaked on TCBS plus 0.2% arabinose to induce the *ccdB* gene and to select for bacteria that had lost the inserted plasmid. Colonies were repurified twice on the same medium, and PCR and sequencing were used on colonies to check for the presence or absence of the target genes. To create the strain ZJ_T/over*csrA*-pSCT32 (the wild-type strain ZJ_T, carrying the CsrA expression plasmid pSCT32-over *csrA*), a PCR fragment containing the *csrA* gene and its flanking regions (including the target gene transcript and terminator sequence, excluding its native promoter) was amplified using the primers over-*csrA*_fwd and over-*csrA*_rev. The relaxed plasmid pSCT32 was amplified with linearized primer pairs over_pSCT32_fwd and over_pSCT32_rev, and the fragment was then inserted into plasmid pSCT32 with a one-step cloning kit (Sangon Biotech, Shanghai, China) to obtain a recombinant plasmid, which was transformed into GEB883-competent cells. The recombinant plasmid pSCT32-over *csrA* was transferred into *V. alginolyticus* ZJ-T by conjugation. The resulting strain ZJ_T/over-*csrA*-pSCT32 was confirmed by PCR analysis and sequencing (Appendix A).

### 2.3. Motility

Overnight cell cultures of ZJ_T, ZJ_T- *csrA.*R6H, and ZJ_T/over *csrA*-pSCT32 were adjusted to OD_600_ = 5.0 in LBS medium. Cultures of 5 μL were spotted on the swimming motility agar plates (0.3% agar) and swarming motility agar plates (1.5% agar). The plates were placed at 30 °C for 16 h. Relative rates of motility were then determined by measuring the diameter of growth. Statistical significance was assayed by the one-way ANOVA LDS method (* *p* < 0.05, ** *p* < 0.01).

A 0.1 to 10-ml pipette tip was used to stab a single colony into the motility agar. The motility medium was T medium supplemented with NRES, and contained 0.3% agar. The inoculated plates were placed upside-down at 37°C for 24 h. Relative rates of motility were then determined by measuring the diameter of growth

### 2.4. Growth Measurement

Bacterial strains were grown overnight in LBS medium at 30 °C with shaking at 200 rpm. To investigate the effect of CsrA on growth in rich medium, each culture was brought to OD_600_ = 1.0 using fresh LBS and then diluted into fresh LBS (1:1000). To investigate the effects of various carbon sources on growth, M63 was modified by replacing D-glucose with 0.4% (*w*/*v*) of different carbohydrates (D-maltose, D-trehalose, D-fructose, D-sucrose, and glycerol). To probe the effect of amino acid(s) on growth, D-glucose and (NH_4_)_2_SO_4_ in M63 were left out and replaced by the amino acids L-alanine (150 mM), L-threonine (50 mM), L-proline (50 mM), and L-serine (50 mM) as carbon and nitrogen sources. Minimal medium assays were carried out as follows: Overnight cultures were collected by centrifugation and washed once with M63 without D-glucose and (NH_4_)_2_SO_4_. Then, they were individually resuspended in the modified M63 to the same cell density. Cultures (3 replicates in each case) were then incubated at 30 °C with continuous shaking at 200 rpm in 96 well plates. OD_600_ was measured at regular time intervals using the Multiskan Ascent plate reader (Thermo Fisher Scientific, Waltham, MA, USA).

### 2.5. RNA Isolation and Quantitative Reverse Transcription PCR (qRT-PCR) Analysis

LBS cultures from single colonies were grown overnight and then diluted 1:1000 in LBS medium and grown to the mid-log phase (optical density at 600 nm [OD_600_] of approximately 0.6), and 1 × 10^9^ bacterial cells were collected. Total RNA was isolated using the TransZol Up Plus RNA Kit (TransGen Biotech, Beijing, China). A total of 1 μg of RNA was reverse-transcribed using a PrimeScript^TM^ RT reagent Kit with the gDNA Eraser, and quantitative PCR (qPCR) was performed with TB Green^®^ Premix Ex Taq™ II (Takara Bio Inc., Kusatsu City, Japan). The relative expression of genes was detected by qPCR using gene-specific primers (Appendix A), and 16S rRNA was used as an internal reference, unless otherwise specified. An Thermal Cycler Dice Real Time System III (Takara Bio Inc.) instrument was used for qPCR with the following parameters: for the holding stage, 95 °C for 30 s; for the two step PCR stage, 95 °C for 5 s and 60 °C for 30 s (repeated 40 times) with the fluorescence recorded at 60 °C; and for the melting curve stage, 90 °C for 15 s, 60 °C for 30 s, and then 95 °C for 15 s with the fluorescence recorded every 0.05 s. Relative levels were calculated using the threshold cycle (ΔΔ*C_T_*) method [29] and normalized to the wild type ZJ-T value. Each reaction only produced one melting curve, indicating that only one target had been amplified during the qPCR reaction. Measurements were done in triplicate. Statistical significance was determined by the one-way ANOVA LDS method (* *p* < 0.05, ** *p* < 0.01).

### 2.6. RNA Stability Measurement

Overnight cultures from a single colony were diluted 1:1000 into LB medium plus 2.5% NaCl (LBS). Cultures were grown to the early-log phase (OD_600_ = 0.5~0.6), and 200 μg/mL of rifampin was added to the culture to stop transcription. Cells were harvested immediately (t = 0) and at 4, 8, 16, and 64 min following the addition of rifampin and were put directly into liquid nitrogen. RNA was then purified from the samples as described above and used to generate cDNA. Those genes, along with control 16S rRNA, were detected by qRT-PCR. The percentage of each of the RNAs remaining at each time point was calculated relative to t = 0 (100%). Given that the percentage remaining of RNA at 4 min after rifampin addition was generally lower than 50%, this point was chosen to represent RNA stability. For the statistical analyses, the one-way ANOVA LDS method was used to compare the rates of decay.

## 3. Results

### 3.1. Construction of *csrA* Pointmutant and Overexpression Strain

In order to examine the role of CsrA in *V. alginolyticus*, we first made numerous attempts to construct an in-frame deletion of *csrA* in the ZJ-T strain. However, none of these attempts succeeded. It has been reported that severe growth defects make it impossible to make an in-frame deletion of *csrA* in several other Gram-negative bacterial species, including *V. cholerae* [21]. Therefore, we constructed a *csrA* point mutation where the sixth codon of CGC (encoding arginine) was replaced by CAC (encoding histidine), which has been shown to dramatically decrease CsrA activity in *V. cholerae* [21]. The mutant was named ZJ-T-*csrA*.R6H. In addition, an overexpression strain ZJ_T/over *csrA* -pSCT32 was constructed by introducing a multiple copy plasmid pSCT32 that harbors a *csrA* gene with the tac promoter. Quantitative RT-PCR was performed to examine the relative expression of *csrA* in WT, ZJ-T-*csrA*.R6H, and ZJ_T/over *csrA* -pSCT32. The results showed that *csrA* is upregulated in ZJ_T/over *csrA* -pSCT32, but not in ZJ-T-*csrA*.R6H (Figure 1), which is not surprising since CsrA has been shown to regulate its own translation rather than transcription.

### 3.2. CsrA Positively Regulates Swarming but Not Swimming Motility in V. alginolyticus

Most *Vibrio* species possess two distinct sets of flagella: the polar flagellum, an organ that drives swimming motility in liquid or semi-solid surfaces, and the lateral flagellum, which drives swarming motility on solid surfaces [30,31,32]. To investigate the role of CsrA in the regulation of motility, 0.3% and 1.5% agar plates were used to evaluate the swimming and swarming motility, respectively. As shown in Figure 2A, the *csrA* mutant was not motile on the 1.5% agar plate. This was in contrast with the wild type and overexpression strains which formed large and similar cell motile zones outwards of the spots. However, on 0.3% agar plates, the strains showed no significant differences from each other (Figure 2A). This indicated that *csrA* is required for the swarming motility of *V. alginolyticus,* which may be mediated by the modulation of lateral-flagella-associated genes. The expression levels of genes encoding lateral flagellin (*lafA*), flagellar basal-body rod protein (*flgC1*), lateral flagellar basal-body rod protein C (*flgC2*), and polar flagellin (*flaC2*) were determined when the bacteria were cultured on 1.5% agar plates. As shown in Figure 2B, *lafA* and *flgC2*, which are required for lateral flagellar biosynthesis, were downregulated by more than 2-fold (*p* < 0.05) in the *csrA* mutant, compared with in the wild-type and *csrA* overexpression strains, while *flgC1* and *flaC2*, which are involved in polar flagella biosynthesis, showed no difference between strains. Taken together, these results may suggest that CsrA regulates the swarming motility by positively affecting the transcript abundance of the genes related to lateral flagella biosynthesis in *V. alginolyticus*.

### 3.3. CsrA Is Extensively Involved in Carbon and Nitrogen Metabolism

To determine the effect of *csrA* on the carbon and nitrogen metabolism of *V. alginolyticus*, the growth characteristics of ZJ_T, ZJ_T-*csrA*R6H and ZJ_T/over*csrA*-pSCT32 were compared in media containing different carbon and nitrogen sources. As shown in Figure 3, there was no significant difference between ZJ_T and ZJ_T-*csrA*R6H when the cells were grown in rich medium LBS, while ZJ_T/over*csrA*-pSCT32 showed a decrease in optical density during the stationary phase. This suggests that excessive CsrA mitigates cellular resistance to the unfavorable environment of the stationary phase, which is characterized by a low pH, low oxygen, and a shortage of carbon/nitrogen sources.

When the cells were grown in minimum media M63 supplemented with NH_4_^+^ ((NH_4_)_2_SO_4_) as the sole nitrogen source as well as glycerol, glucose, sucrose, trehalose, or maltose as sole carbon source, ZJ_T/over*csrA*-pSCT32 exhibited a prolonged lag phase compared with the other two strains. However, the growth kinetics at the log phase were similar. This suggests that the overexpression of *csrA* may slow down the uptake and accumulation of nutrients, which are the major cellular processes that occur during the lag phase, but not cell division. Interestingly, both the *csrA* mutant and overexpression strains showed decreased maximum cell densities during the stationary phase when glycerol was the sole carbon source. Under these conditions, the cells must reorganize two major biophysical processes, the catabolism of carbohydrates and the anabolism of amino acids de novo, to provide sufficient nucleotides and amino acids, which are the building blocks of macro-molecules required for fast growth. Given that the carbohydrates tested were taken up through different pathways, it can be hypothesized that CsrA may primarily regulate amino acid anabolism.

To investigate the impact of CsrA on the metabolism of amino acids, bacterial growth was measured in M63 supplemented with serine, threonine, alanine, or proline. As shown in Figure 3, in the medium containing serine, ZJ_T-*csrA*R6H had the longest lag phase, followed by ZJ_T/over*csrA*-pSCT32, while ZJ_T had the shortest lag phase, but all strains exhibited similar growth rates during the log phase and had maximum cell density during the stationary phase. When cultured in medium containing threonine, ZJ_T-*csrA*R6H and ZJ_T/over*csrA*-pSCT32 showed similar prolonged lag phases compared with ZJ_T, but ZJ_T-*csrA*R6H underwent a second growth phase when the other two reached the steady stationary phase, leading to a higher maximum cell density. In M63 with alanine, ZJ_T and ZJ_T-*csrA*R6H exhibited the same growth curve, but ZJ_T/over*csrA*-pSCT32 showed a prolonged lag phase, slow growth rate, and low cell density during the stationary phase. When proline was used as the sole carbon and nitrogen source, ZJ_T-*csrA*R6H did not grow at all, but ZJ_T/over*csrA*-pSCT32 showed a shorter lag phase, a faster growth rate, and a similar cell density during the stationary phase. These results indicate that CsrA may regulate the catabolism of amino acids in an amino-acid-specific manner in *V. alginolyticus*. In the situation tested above, amino acids were presumably taken up into the cell at first, before undergoing oxidative deamination to form their corresponding α-keto acids and ammonia, a process catalyzed by L-amino acid deaminases. The α-keto acids were then utilized for central metabolism, producing ATP, NAD(P)H, and carbon intermediates for the biosynthesis of other amino acids together with ammonia. Therefore, it is possible that *csrA* functions in the importation and/or the degradation of amino acids.

### 3.4. CsrA Has a Significant Impact on the Transcript Abundance of Genes Involved in Central Metabolism

To better understand the mechanism responsible for the phenotypical changes observed above, the expression profiles of genes associated with carbon and nitrogen metabolism were selected (Appendix A) and determined by qRT-PCR. The results are presented and discussed in categories based on their functions.

The expression profiles of *fbp*, *pfkA*, *fbaA*, *eno*, and *glgA* were selected and measured (Figure 4). Except for *fbp* and *glgA*, the genes showed similar expression profiles in all strains, indicating that CsrA does not affect the transcription and/or mRNA stability of the genes involved in glycolysis. However, *fbp*, which encodes fructose-1,6-bisphosphatase, and *glgA*, which encodes glycogen synthase, were upregulated by more than 2-fold in ZJ_T-csrAR6H compared with ZJ-T. This suggests that CsrA may inhibit gluconeogenesis and glycogen biosynthesis.

Compared with ZJ_T, ZJ_T-*csrA*R6H displayed 3–4 fold increases in *pdhA* and *aceE*. These genes have been reported to encode two different sets of pyruvate dehydrogenase (acetyl-transferring) E1 component subunit alpha. However, *aceE* was also upregulated 3-fold in ZJ_T/over*csrA*-pSCT32. This implies that *csrA* may negatively regulate the reaction from PEP to pyruvate, leading to the restriction of carbon flux into the TCA cycle. The genes *cs*, *acnB*, *sucA*, *sucC*, and *sdhA* which are responsible for the TCA cycle were all upregulated in ZJ_T-*csrA*R6H by over 5-fold. In addition, *ppc*, *pckA*, and *ppsA*, which are involved in the conversion of oxaloacetate, PEP, and pyruvate increased by more than 2-fold in ZJ_T-csrAR6H. These results indicate that CsrA negatively regulates the pyruvate and TCA cycles at multiple points.

As shown above, *csrA* is apparently involved in the regulation of serine, threonine, alanine, and proline metabolism. To determine its effects on serine metabolism, the expression profiles of *sdaA*, *sdaB*, and *sdaC*, which encode the serine transporter (*sdaC*) and two serine deaminases (*sdaA* and *sdaB*) were measured. As shown in Figure 5, the expression of *sdaC* was at the same level regardless of whether *csrA* was tuned up or down, but *sdaA* was downregulated by 2-fold and *sdaB* was upregulated by 3-fold in the *csrA* mutant. However, no significant difference was found between ZJ-T and ZJ_T/over*csrA*-pSCT32. For alanine metabolism, the expression profiles of *ald* and *avtA*, which are involved in the utilization and conversion of alanine, were quantified. *ald*, which encodes alanine dehydrogenase, was increased by more than 8-fold in the *csrA* mutant compared with the wild-type strain, while *avtA* showed no difference between strains. *proV* and *proB* are involved in proline transport and utilization. When *csrA* was altered, *ProV* was upregulated by 7-fold, but *proB* remained at a similar expression level. In addition, *gltB* and *gltD,* encoding for glutamate synthase, which is central to the biosynthesis of amino acids, were upregulated by over 8-fold in *csrA* mutant ZJ_T-*csrA*R6H.

In summary, CsrA extensively represses the transcript abundance of genes involved in central metabolism, which may have profound effects on pathways such as gluconeogenesis, glycogen biosynthesis, the pyruvate cycle, and the TCA cycle (Figure 6), as well as amino acid uptake and utilization (Figure 7).

### 3.5. CsrA Changes the Transcipt Abundance of Genes Involved in Central Metabolism Likely by Alternation of Their mRNA Stability

CsrA is an RNA-binding protein that acts on single-stranded RNA molecules, so it is unlikely that it alters the abundance of transcripts by working on their transcription. A possible reason is that binding to CsrA may change the half-life of the target mRNA. To test this hypothesis, we measured the RNA stability of *serA*, *pdhA*, *acnB*, *sdhA*, *ppc*, *sucC*, *ald*, *gltD*, and *cs*. As shown in Figure 8, less than half of the transcripts of all genes remained after 4 min of rifampicin addition, but the quantity of residuals showed great variation among genes, ranging from 50% to less than 10%, indicating that the half-life of most mRNAs in *V.alginolyticus* may be less than 4 min. In addition, changes in the *csrA* abundance significantly altered the stability of the mRNA. For example, after 4 min, only 10.1% of *serA* transcripts remained in the wild type, while 37.8% remained in the *csrA* mutant. Similar results were obtained for *pdhA* (20.9% vs. 51.9%), *acnB* (12.8% vs. 45.5%), *ppC* (5.3% vs. 25.7%), and *cs* (20% vs. 35.5%). Meanwhile, the numbers of residual transcripts of *ald* (49.5% vs. 14.6%), *cs* (20.0% vs. 8.7%), *gltD* (23.2% vs. 3.1%), *sdhA* (22.2% vs. 2.4%), and *sucC* (34.2% vs. 4.2%) significantly reduced in the *csrA* overexpressed strain. These results suggest that binding to CsrA may accelerate target mRNA decay, leading to a decrease in mRNA abundance.

The consensus sequence of CsrA binding is ARGGAN, in which GGA is mandatory. To identify the possible CsrA binding site(s) in the genes measured above, we retrieved sequences ranging from −50 to +50 of the initiation codon and searched for the GGA motifs. As shown in Appendix A and Figure 9, out of 33 genes, only five genes (*avtA*, *proB*, *sdaC*, *pfkA* and *sdhA*) were not found to contain a GGA motif. Except for *acnB*, *tdh*, *pfkA*, *flgC2* and *ppsA*, the genes (22) enclosed one or two GGA motifs located within the range of −20 to the initiation codon, which covers the ribosome binding sites. Nine genes showed no difference in the number of transcripts when *csrA* was mutated, among which four had no GGA motif(s) and four had only one in their sequences. Taken together, these results imply a strong correlation between the potential *csrA* binding sites and repression of CsrA.

## 4. Discussion

It has been widely reported that CsrA is a global regulator of *gamma*-proteobacteria that affects essential biochemical and biophysical processes. In *vibrio* species, it has been reported to modulate biofilm formation, quorum sensing, virulence, and amino acid metabolism [21,33,34,35]. In this study, the manipulation of CsrA resulted in variations in swarming motility and carbon and nitrogen metabolism but did not alter biofilm formation. Correspondingly, extensive changes in the expression of genes associated with these phenotypes were also observed. These results suggest that CsrA is a global regulator that modulates motility and central metabolism in *V. alginolyticus*.

The genome of *V. alginolyticus* has two sets of genes, which encode a polar flagellum and numerous lateral flagella, respectively. Expression of the polar flagellum is induced in liquid, which enables cells to swim in a liquid environment, while lateral flagella are synthesized only when cells are grown on solid surfaces or in a high-viscosity environment [30,31,32]. In this study, the mutation of *csrA* resulted in a lower swarming motility but no difference in swimming motility, which corresponded to the reduced expression of lateral flagellar genes, i.e., *lafA*, *flgC2* in the *csrA* mutant, indicating that CsrA may positively regulate swarming motility by upregulating the expression of lateral flagellar genes in *V. alginolyticus*. In *E. coli*, CsrA was reported to enhance bacterial motility by binding to the 5′ UTR of *flhDC* transcript, which encodes an activator of flagella biosynthesis and chemotaxis, thus protecting it from RNase E-mediated cleavage [16,36]. Although no experiment has been carried out to study the regulatory effect of CsrA on motility in *Vibrio* species, RNA-Seq revealed that over half of all annotated flagellum-dependent motility genes, including *rpoN* and *flrC*, are significantly downregulated in the *csrA* mutant in *V. cholera*. CsrA was shown to bind directly to both *rpoN* and *flrC* mRNA, which encode master activators of flagella synthesis and chemotaxis in vivo [23]. However, unlike *V. alginolyticus*, *V. cholerae* possesses only one polar flagellum, so the mechanisms underlying their motility regulation may be very different, making it necessary to further explore the role of CsrA in motility regulation in *V. alginolyticus*.

*Vibrio* is widely distributed in the marine environment and has a very fast growth rate [37], suggesting it has the ability to effectively uptake, utilize, and convert nutrients. In this study, the overexpression of *csrA* led to a decrease in the maximum cell density during the stationary phase in rich medium, which may suggest its involvement in the stress response at the stationary phase. It has been reported that *rpoS*, which encodes the major *sigma* factor during the stationary phase, is negatively regulated by CsrA in *V. cholerae* [23]. Interestingly, a prolonged lag phase was observed when cells with overexpressed *csrA* grew in minimum media supplemented with various carbohydrates as the sole carbon source and ammonium as the sole nitrogen source. However, no difference in growth rate was shown during the exponential phase. This indicated that *csrA* may repress major biochemical processes during the lag phase, a distinctive growth phase in bacterial culture. The duration of lag phase depends on a lot of factors, but it is mainly influenced by the inoculum history [38]. When the inoculum has more viable cells during the last stationary phase, the lag period is shorter, which may be one of the reasons why the overproduction of CsrA resulted in prolongation of the lag phase.

In addition, the cells do not replicate during the lag phase, but active cellular processes are undertaken to produce cellular components that are prepared for exponential growth [39,40]. Therefore, optimizing primary metabolism to establish maxim carbon flow in response to available nutrients is the significant characteristic of the lag phase. Yamamotoya et al. demonstrated that intracellular glycogen, instead of glucose, imported from the environment is the primary source of central metabolism during the lag phase [41]. This implies that the speed of carbohydrate uptake may not be the cause of the prolonged lag phase in the *csrA* overexpression strain of *V. alginiolyticus*. Under the conditions measured above, active metabolic pathways of glycolysis, the pyruvate cycle, and amino acid biosynthesis de novo must be required for the growth of bacteria. It is reasonable to speculate that the repression of one or more of these genes would result in a prolonged lag phase. Indeed, the expression profiles of genes associated with these pathways provide preliminary evidence in support of this notion. Based on the abundance of transcripts, the mutation of *csrA* may significantly promote gluconeogenesis, glycogen synthesis, and the pyruvate cycle, but not glycolysis (Figure 6).

CsrA is essential for carbohydrate metabolism in *E. coli* and other gram-negative bacteria. It has long been reported that CsrA promotes glycolysis but inhibits gluconeogenesis and glycogen synthesis [42]. The interconversion of fructose 6-phosphate and fructose 1,6-diphosphate, catalyzed by 6-phosphofructokinase (PfkAB) and Fructose-1,6-bisphosphatase (Fbp), respectively, is the limiting step of glycolysis and gluconeogenesis. Thus, these processes are the critical regulatory checkpoints of central metabolism regulation. In *E. coli*, CsrA positively regulates *pfkA* at the levels of both mRNA stability and translational efficiency, leading to increased carbon flux into glycolysis [43]. However, in *V. cholerae*, CsrA has been shown to be a repressor rather than an activator of *pfkA* that only works during the stationary phase [23]. In this study, alternation of *csrA* had no effect on the expression of *pfkA* at the early-log phase, and it was not found to contain a potential CsrA binding site in the sequence between −50 and +50 nt around the initiation codon (Figure 9). In addition, other enzymes involved in glycolysis/gluconeogenesis displayed no difference in the abundance of transcripts, indicating that CsrA may not participate in the regulation of glycolysis in *V. alginolyticus* at the early-log phase. However, CsrA was shown to repress gluconeogenesis and glycogen synthesis through the inhibition of *fbp* and *glgA* during the early exponential growing period*,* which coincides with the results of other studies on *E. coli* and *V. cholerae* etc. [42].

Carbon flux from the PEP-pyruvate-AcCoA to TCA cycle is crucial for the integration of carbohydrates with energy production and amino acid metabolism. Recent studies have suggested a new pathway of central metabolism called the pyruvate cycle, or P cycle, which merges the PEP-pyruvate-AcCoA pathway with the TCA cycle [44]. In this study, CsrA inhibited the expression of genes involved in the P cycle extensively, a feature that has also observed in other studies [45]. It appears that CsrA-mediated multipoint inhibition of the P-cycle is a conserved feature of central metabolic regulation, so it can be expected that the overexpression of *csrA* will lead to a deceleration of energy production, reducing the concentration of carbon intermediates undergo amino acid biosynthesis de novo. This may be one of the reasons why the *csrA* overexpressed strain showed a prolonged lag phase when grown in the media with ammonium as the sole nitrogen source.

Amino acids are critical components that not only act as building blocks for protein synthesis but also serve as both carbon and nitrogen sources for growth. Here, we found that the alternation of CsrA significantly changed growth rates, the duration of the lag phase, or the maximum cell density of *V. alginolyticus* when serine, alanine, threonine, or proline served as the sole carbon and nitrogen source. Q-RTPCR has also shown that genes encoding for specific transporters and enzymes for their degradation had altered expression levels in the *csrA* mutant. It is not well known how CsrA regulates amino acid catabolism, but Ina et al. [46] suggested that CsrA regulates a switch from amino acid to glycerolipid metabolism in *Legionella pneumophila* and in *V. cholerae*, extensive changes in transcripts involved in amino acid uptake and degradation were found when *csrA* was impaired [23]. This may suggest that CsrA is extensively involved in the regulation of amino acid transport and catabolism by changing the expression levels of relevant genes.

In *V. alginolyticus*, the levels of a wide range of gene transcripts were significantly altered in the *csrA* mutant, which is in agreement with the results of other studies [42,47,48,49]. However, as an RNA binding protein, CsrA is not expected to function as a transcriptional regulator that affects the transcription of target genes. Heidi et al. 2021 suggested that this is because CsrA is able to regulate other global regulators, such as *rpoS/rpoE* [23]. Another plausible explanation is that binding to CsrA may affect mRNA stability. In this study, we found that impairment or overexpression of *csrA* resulted in longer or shorter half-lives for dozens of mRNAs, suggesting that binding to CsrA may accelerate mRNA decay, leading to the destabilization of target mRNAs in *V. alginolyticus*, which is in agreement with the overall repression of RNA abundance. However, in *E. coli*, CsrA was suggested to be a global positive regulator of mRNA stability, although possible mechanism(s) behind were not addressed [50].

In summary, this study provides the first insights into the regulatory role of CsrA in *V. alginolyticus.* Our results highlight the importance of post-transcriptional controls mediated by CsrA that likely contribute to the fast growth and adaptation of this bacteria in response to a changing environment. However, since it is a post-transcriptional global regulator, transcriptomic and proteomic analyses are needed to further reveal its role in *V. alginolyticus*.

## Figures and Tables

**Figure 1 microorganisms-09-02383-f001:**
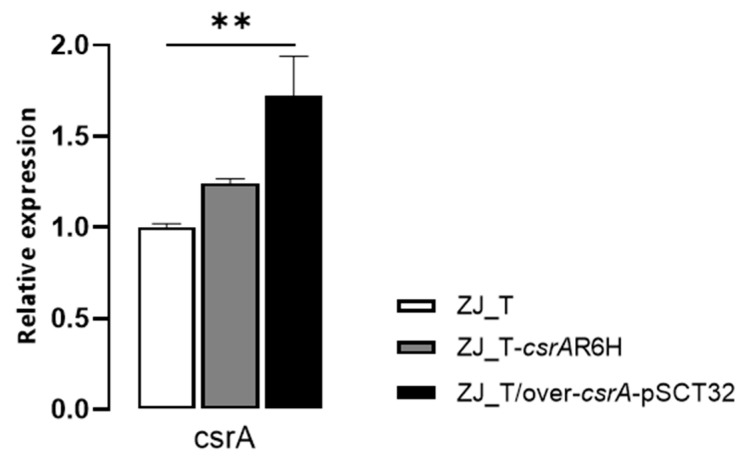
The *csrA* mutant did not affect transcription. The relative expression of *csrA* behaved at the wildtype level in terms of expression in the *csrA* mutant, while it was upregulated in the overexpression strain (One-way ANOVA, FDR ** *p* < 0.01). The levels of *csrA* were normalized to the internal control *recA* level. Error bars indicate standard deviations.

**Figure 2 microorganisms-09-02383-f002:**
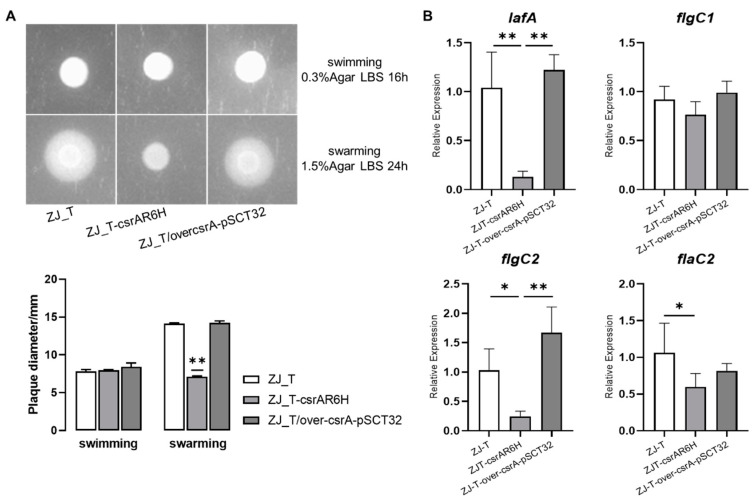
CsrA positively regulates swarming but not swimming motility in *V. alginolyticus*. (**A**) The ability of ZJ_T, ZJ_T-*csrA*R6H, ZJ_T/over*csrA*-pSCT32 to swim or swarm through 0.3% and 1.5% LBS agar plates, which was observed with a BIO-RAD Gel Doc^TM^ XR^+^ imager. The presence of the *csrA* mutant did not change the swimming ability, but it decreased the swarming ability. The bars represent the mean of three biological replicates, the error bars are the standard deviations, and the *p* values were calculated using one-way ANOVA (* *p* < 0.05, ** *p* < 0.01) comparing ZJ_T-*csrA*R6H to both the wild type and the overexpression strain. (**B**) Relative expression levels of lateral flagellin (*lafA*), flagellar basal-body rod protein (*flgC1*), flagellar basal-body rod protein C (*flgC2*), flagellin (*flaC2*). *lafA*, *flgC2*, *flaC2* were significantly downregulated and displayed changes of greater than 2-fold (Baggerley’s test, FDR *p* value of <0.05) in the expression in the *csrA* mutant compared with the wild-type strain, while *flgC1* was downregulated but not significantly. Error bars indicate standard deviations.

**Figure 3 microorganisms-09-02383-f003:**
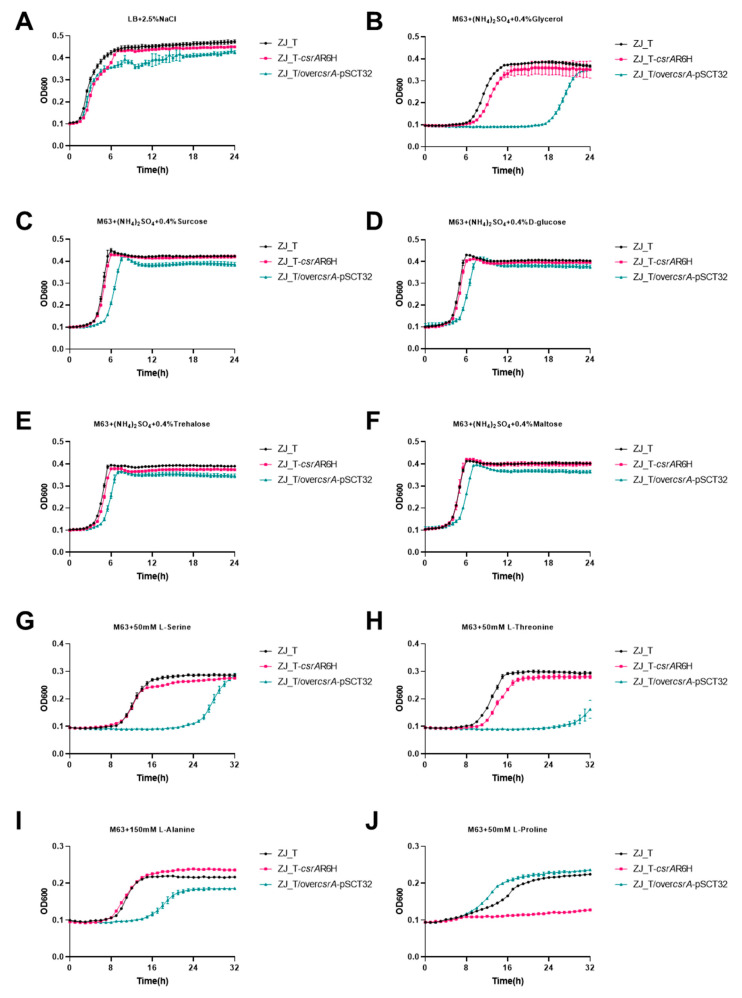
CsrA is extensively involved in carbon and nitrogen metabolism. Growth curves of the wild type, *csrA* point mutation, and overexpression strains grown in LB + 2.5% NaCl rich medium (**A**), minimal medium M63 containing (NH_4_)_2_SO_4_ and 0.4% glycerol (**B**), M63 plus (NH_4_)_2_SO_4_ plus sucrose (**C**), M63 plus (NH_4_)_2_SO_4_ plus glucose (**D**), M63 plus (NH_4_)_2_SO_4_ plus trehalose (**E**), and M63 plus (NH_4_)_2_SO_4_ plus maltose (**F**) and in M63 which D-glucose and (NH_4_)_2_SO_4_ were left out and replaced by the amino acids L-Serine (50 mM) (**G**), L-Threonine (50 mM) (**H**), L-Alanine (150 mM) (**I**), and L-Proline (50 mM) (**J**) as the sole carbon and nitrogen sources. For growth curves, three biological replicates are shown as points with their average values connected by lines. Error bars indicate the standard error of mean (SEM). ZJ_T-*csrA*R6H exhibited wild-type growth in rich medium and M63 modified by 0.4% (*w*/*v*) of different carbohydrates, but overexpression strains showed longer lag phases for all carbon sources tested. When M63 was supplemented with serine, the lag phase of the other strains was prolonged compared with that of the wild type, but the total growth was the same in general. When M63 was supplemented with Threonine, the lag phase of the *csrA* mutant and overexpression strains was prolonged like in serine, but the *csrA* mutant displayed an increased growth yield. When M63 was supplemented with Alanine, *csrA*.R6H exhibited wild-type growth, but the duration of the lag phase and final biomass of the overexpression strain decreased obviously. In addition, When M63 was supplemented with Proline, the mutation of *csrA* affected the ability of *V. alginolyticus* to grow on L-Proline.

**Figure 4 microorganisms-09-02383-f004:**
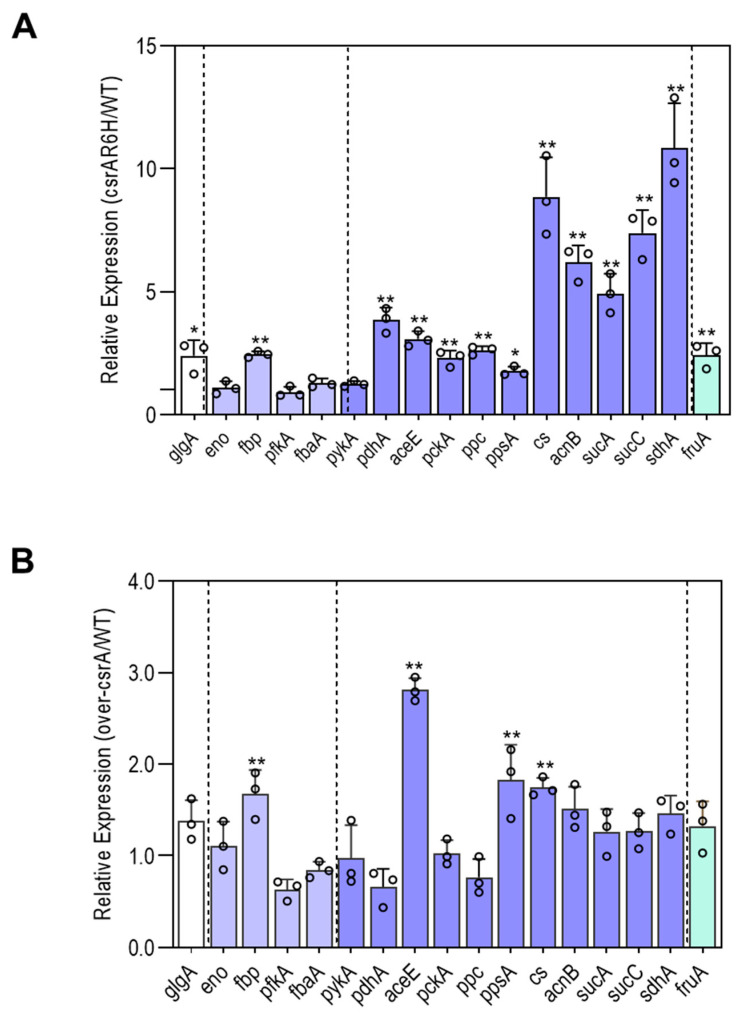
CsrA has a significant impact on the transcript abundance of genes involved in central metabolism. The color code matches the glucogenesis pathway (*glgA*), glycolysis pathway (*eno*, *fbp*, *pfkA*, *fbaA*), pyruvate cycle, TCA cycle (*pykA*, *aceE*, *pckA*, *pdhA*, *ppc*, *ppsA*, *cs*, *acnB*, *sucA*, *sucC*, *sdhA*), and nutrient transporter (*fruA*), respectively. (**A**) The relative expression of the *csrA*.R6H strain divided by that of the WT. (**B**) The relative expression of the over-*csrA* strain divided by that of the WT. The relative expression level of a set of genes was obtained for each strain from the integration of three biological replicates (see Materials and Methods for details). Data are presented as the mean ± SD; * *p* < 0.05, ** *p* < 0.01 by one-way ANOVA with the FDR post hoc test comparing ZJ_T-*csrA*R6H and ZJ_T/over-*csrA*-pSCT32 to the wild type.

**Figure 5 microorganisms-09-02383-f005:**
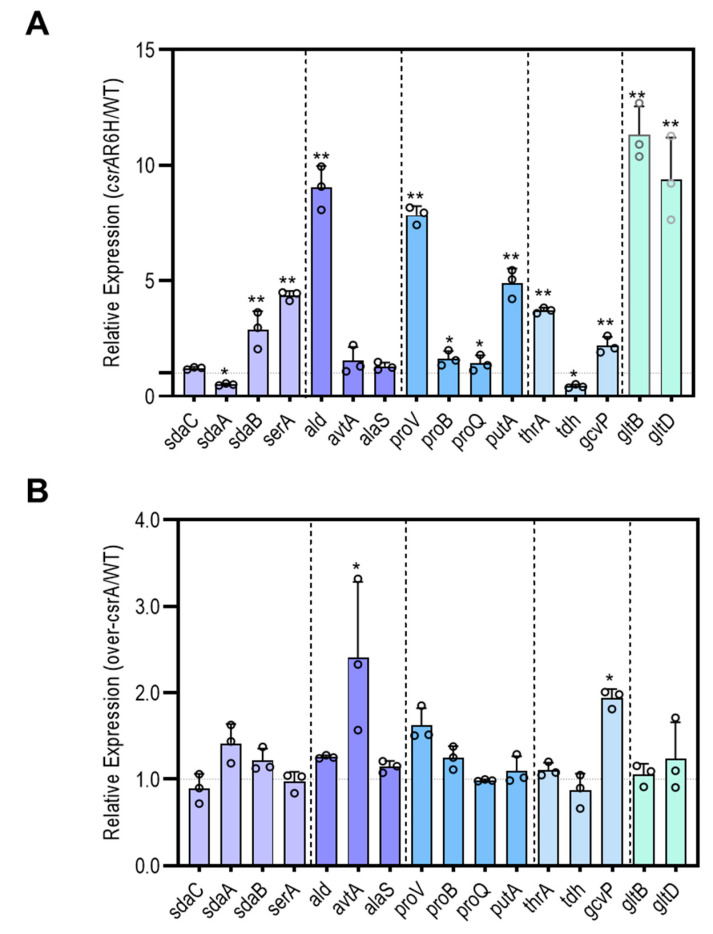
CsrA affects the expression of genes related to amino acid metabolism. The color code matches the genes involved in amino acid metabolism, serine metabolism (*serA, sdaC*, *sdaB*, *sdaA*), alanine metabolism (*ald*, *avtA*, *alaS*), proline metabolism (*proV*, *proQ*, *proB*, *putA*), threonine (*thrA*, *tdh*, *gcvP*), and glutamine metabolism (*gltD*, *gltB*). (**A**) The relative expression of the *csrA*.R6H strain divided by that of the WT. (**B**) The relative expression of the over-*csrA* strain divided by that of the WT. Data are presented as the mean ± SD. * *p* < 0.05, ** *p* < 0.01 by one-way ANOVA with the FDR post hoc test, comparing the ZJ_T-*csrA*R6H and overexpression strain to the wild type.

**Figure 6 microorganisms-09-02383-f006:**
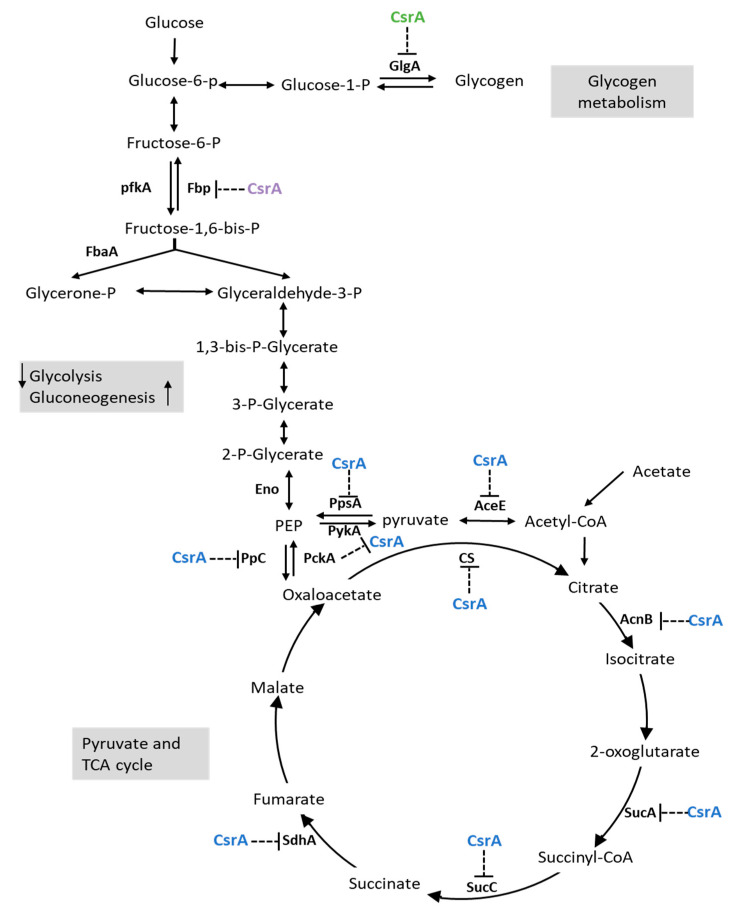
Post-transcriptional regulation of the central carbon metabolic pathways in *V. alginolyticus*. Effects of the RNA-binding protein CsrA on the synthesis of enzymes involved in glycogen metabolism, glycolysis, gluconeogenesis, and the pyruvate and TCA cycles. Green, blue, and purple represent glycogen metabolism, glycolysis and gluconeogenesis, and the pyruvate and TCA cycles respectively. GlgA, glycogen synthase; PfkA, 6-phosphofructokinase; FbaA, fructose-bisphosphate aldolase; FBP, Fructose-1,6-bisphosphatase; Eno, phosphopyruvate hydratase; PykA, pyruvate kinase II; AceE, pyruvate dehydrogenase (acetyl-transferring); AcnB, bifunctional aconitate hydratase 2/2-methylisocitrate dehydratase; CS, citrate (Si)-synthase; SucA, 2-oxoglutarate dehydrogenase; SucC, succinyl-CoA synthetase, SdhA, succinate dehydrogenase catalytic; PckA, phosphoenolpyruvate carboxykinase; PpC, phosphoenolpyruvate carboxylase; PpsA, phosphoenolpyruvate synthase; FruA, PTS fructose transporter subunit IIBC.

**Figure 7 microorganisms-09-02383-f007:**
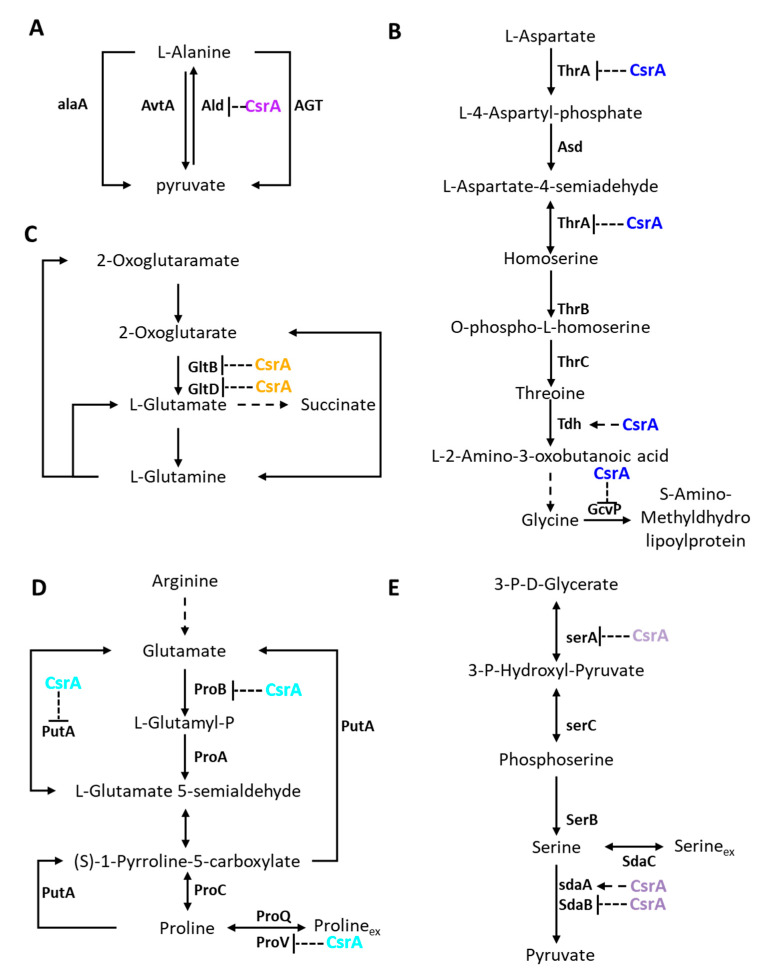
Schematic of metabolic reactions and relevant enzymes involved in amino acid metabolism. Effects of the RNA-binding protein CsrA on the synthesis of enzymes involved in alanine (**A**), glutamine (**B**), threonine (**C**), proline (**D**), and serine (**E**) biosynthesis and utilization. Purple, blue, orange, cyan, and dark goldenrod represent alanine, glutamine, threonine, proline, and serine biosynthesis and utilization respectively. Ald, alanine dehydrogenase; AlaA, alanine-synthesizing transaminase; AvtA, valine--pyruvate transaminase; AGT, pyruvate transaminase; GltD, glutamate synthase, GltB, glutamate synthase. ThrA, bifunctional aspartate kinase/homoserine dehydrogenase I; Asd, aspartate-semialdehyde dehydrogenase; ThrB, homoserine kinase; ThrC, threonine synthase; Tdh, L-threonine 3-dehydrogenase; GcvP, glycine dehydrogenase; ProV, ABC superfamily (glycine/betaine/proline transport protein); ProQ, RNA chaperone; ProB, gamma-glutamyl kinase; PutA, bifunctional proline dehydrogenase/L-glutamate gamma-semialdehyde dehydrogenase; ProA, glutamate-5-semialdehyde dehydrogenase; ProC, pyrroline-5-carboxylate reductase; SdaC, serine transporter; SdaA, L-serine dehydratase 1; SdaB, L-serine ammonia-lyase; SerA, D-3-phosphoglycerate dehydrogenase; SerB, phosphoserine phosphatase; SerC, phosphoserine aminotransferase.

**Figure 8 microorganisms-09-02383-f008:**
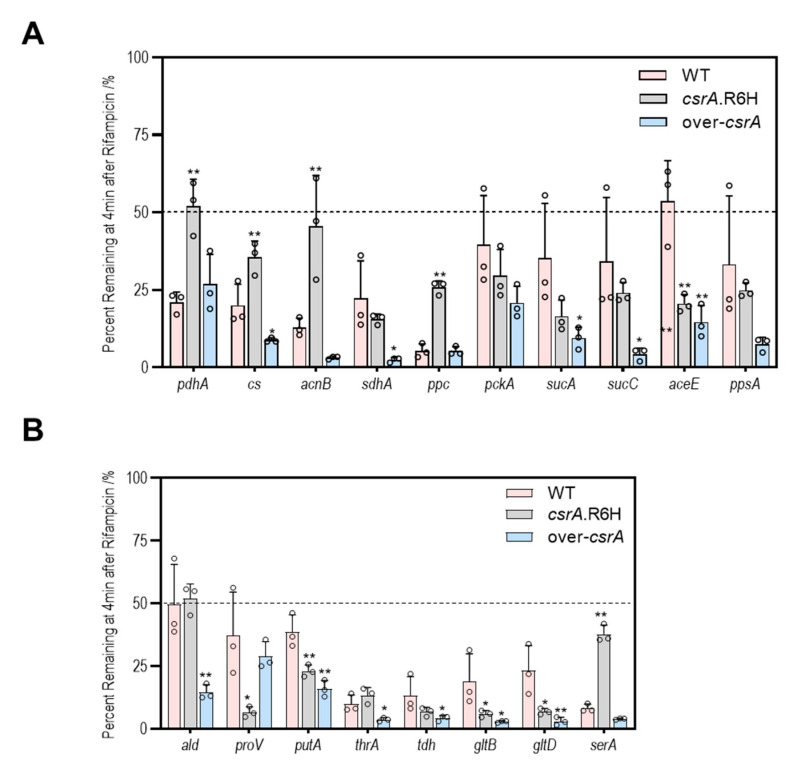
CsrA changes the transcript abundance of genes involved in central metabolism, likely by alternation of their mRNA stability. (**A**) The percent remaining at 4 min after rifampin addition. The rates of decay for *pdhA*, *cs*, *acnB*, *sdhA*, *ppc*, *pckA*, *sucA*, *sucC*, *aceE*, and *ppsA* (which were up-regulated or downregulated significantly and displayed greater than 3-fold changes in expression in the *csrA* mutant or over csrA strain compared with the wild-type strain) was measured by qRT-PCR from wild-type, *csrA*.R6H, or over-*csrA* cells grown to the mid-log phase (OD600 ~0.5) in LBS medium. (**B**) Percent remaining at 4 min after rifampin addition. The rates of decay for *ald*, *proV*, *putA*, *thrA*, *tdh, gltB*, *gltD*, and *serA* (which were up-regulated or downregulated significantly and displayed greater than 2-fold changes in expression in the *csrA* mutant or over csrA strain compared with the wild-type strain) were measured by qRT-PCR from wild-type, *csrA*.R6H, and over-*csrA* cells grown to the mid-log phase (OD600 ~0.5) in LBS medium. Rifampin was added to stop transcription, and cells were harvested immediately (t = 0) and at 4 min after rifampin addition. The displayed values correspond to the percentage of RNA remaining at 4 min after rifampin addition in data showing the degree of RNA decay. The error bars are the standard deviations, and the *p* values were calculated using one-way ANOVA (*p* values: *, <0.05, **, <0.01), comparing the ZJ_T-*csrA*R6H and overexpression strains to the wild type.

**Figure 9 microorganisms-09-02383-f009:**
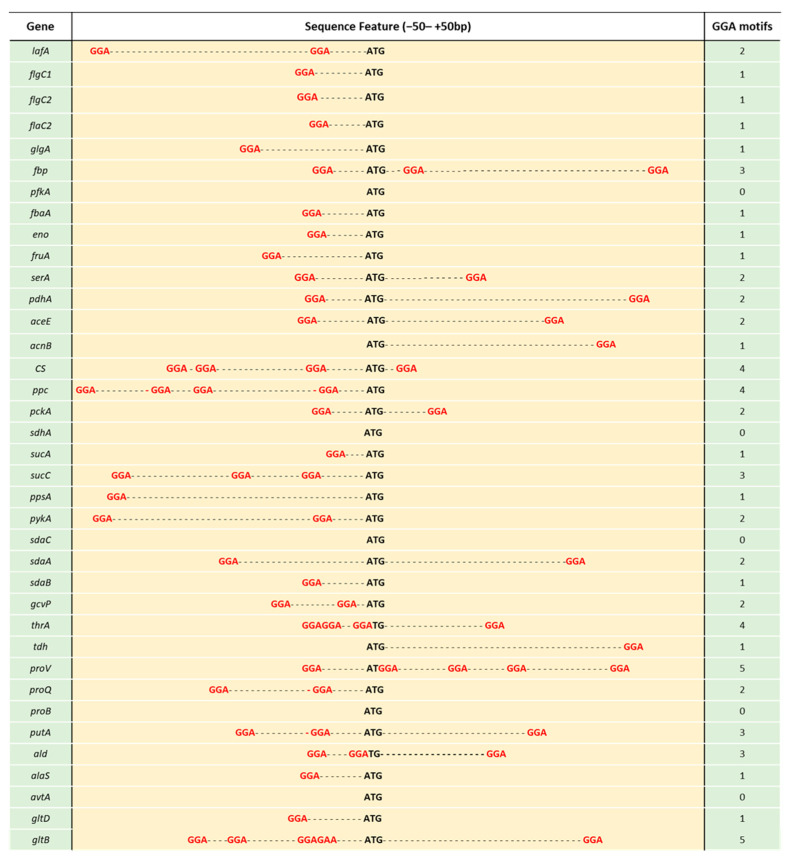
GGA motifs of sequences ranging from −50 to +50 bp of the initiation codon. Bold letters (depicted in black) indicate the start codon and bold letters (depicted in red) indicate GGA motifs. A short horizontal line represents the base of a gene.

**Table 1 microorganisms-09-02383-t001:** Strains and plasmids used in this study.

Strains or Plasmids	Relevant Characteristics	Source
*Vibrio alginolyticus*		
ZJ_T	Ap^r^ (ampicillin-resistant), translucent/smooth variant of wild strain ZJ51; isolated from diseased *Epinephelus coioides* off the Southern China coast	[26]
ZJ_T-*csrA* R6H	Ap^r^; ZJ-T carrying a point mutation that replaces the arginine residue at amino acid position 6 with a histidine (R6H).	This study
ZJ_T/over-*csrA*-pSCT32	Cm^r^; ZJ-T carrying the CsrA expression plasmid pSCT32- over-*csrA*	This study
*E. coli*		
GEB883	WT; E.coli K12 Δ*dapA::ermpir* RP4-2 Δ*recA gyrA462*, *zei*298::Tn10; donor strain for conjugation	[27]
Plasmids		
pSW7848	Cm^r^; suicide vector with an R6K origin, requiring the Pir protein for its replication, and the *ccdB* toxin gene	[24]
pSW7848-CsrAR6H	Cm^r^; pSW7848 containing the mutant allele of CsrA	This study
pSCT32	Cm^r^; expression plasmid with a pBR322 and a f1 origin at the same time and a tac promoter	[28]
pSCT32-over *csrA*	Cm^r^; pSCT32 containing the WT allele of CsrA	This study

Note: Cm^r^ and Ap^r^ indicate chloramphenicol and ampicillin resistance, respectively.

## Data Availability

All datasets generated for this study are included in the article/Appendix A.

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
