# Peer review of "CsrA Regulates Swarming Motility and Carbohydrate and Amino Acid Metabolism in Vibrio alginolyticus"

_microorganisms, 2021, doi:10.3390/microorganisms9112383_

Round 1
Reviewer 1 Report
Summary
The authors based their study on the importance of Vibrio alginolyticus in marine aquaculture industry and its ability to rapid proliferation in abundant organic matter environments. Following the recent advances in knowledge of the implication of the CsrA in V. cholera transition from environment to human host, and the lack of investigations of the role of this regulator in V. alginolyticus, the authors analyzed the role of CsrA in its motility and the central metabolism regulation. The assays were carried out in parallel under reduced activity of CsrA (R6H point mutation) and overexpression conditions (expression of CsrA WT from multicopy plasmid). The authors found that CsrA activates swarming rather than swimming motility by regulating the expression of lateral flagellar associated genes. The authors also detected changes in transcripts of the genes encoding key enzymes from different metabolic pathways using the RT-qPCR technique. Considering the extensive list of gene expression analyzed and in light of the results obtained, the study shows for the first time the role of CsrA as a regulator of various cellular processes in V. alginolyticus.
The rationale followed by the authors in the design and performance of the experiments is accurate to answer the proposed questions. Methods, tools, software and reagents are described in detail and applied statistical types are mentioned along with the significance values. The results are mostly well presented, some minor suggestions are made in the comments below.
General concept comments
The authors are asked to please address the following comments.
- There is a misuse of the word knockdown referring to the experiments done with the strain ZJ_T-csrA R6H (lines 376, 451, 472). Although the point mutation R6H leads to a decrease in CsrA activity, it cannot be considered as knockdown since its expression is not reduced compared to the WT.
- Seems there is an inconsistent nomenclature in Table 1 and line 232 related to promoter from pSCT32 and derivative plasmid pSCT32-over csrA, are they the same?
- Figures 1A and 1B are not described in results nor discussed. The authors might consider to transfer them to a Supplementary Materials section. Along the same lines, Tables 2 and 3 would be more appropriate to include in a Supplementary Materials section.
- Table 4: GGA motifs column seems repetitive to Figure 10, the information related to the genes lafA, flgC1, flgC2, flaC2 from this table could be included in Figure 10. It would be helpful for the readers if authors highlight or specify in Table 4 the genes used for mRNA stability experiments as well, and afterward relate it to the Table 4 in line 196. Significance (csrA.RH6 & WT) column is also redundant since the same information is plotted in Figures 5 and 6.
- The authors mention the rate of decay to describe the RNA stability measurement results (lines 201, 432, 436). However, according to the description in Material and Methods section of the calculations they made and the graphs in Figure 9, seems that the results only represent the percentage of the genes still detectable after the treatment with rifampicin. Could the authors define or clarify the term rate of decay?
- Figures 7 and 8 contain too much information that it is difficult to correlate to the results from Figures 5 and 6. Following the same color codes employed in Figures 5 and 6 would facilitate the visualization of the results in the metabolic pathways’ diagrams.
- In general, thought the whole text, the authors should revise the consistency of the name of the three strains/plasmids used as well as the nomenclature when referring to protein and/or genes.
- Overall, the manuscript is written in an appropriate and understandable way. However, the grammar should be revised in some sentences and others seem incomplete (e.g. lines 61-63, 81-83, 94-103, 174-175, 228-229).
Specific comments
- Reference for plasmid pSW7848 is duplicated, number [24] in line 79 and number [28] in Table 1. Need to delete the [28] and relabel de following references.
- Table 1: revise the column Relevant characteristics of the plasmid pSCT32-over csrA, should be Cmr; pSCT32 containing the WT allele of CsrA?
- Figure 4C, 4D, 4E: in the tittle of the figures should be 0.4% according to the information mentioned in line 160.
- Line 178: add 16S rRNA.
- Table 3: seems the primer sequences for qPCR of the genes csrA and the controls recA and rrsA are missing.
- Line 205: specify Figure 1C.
- Line 247: Figure 3A instead of 1A.
- Figure 4: error bars should be displayed for each time point in each graph. In case they are too small and cannot be observed, this information should be specified in the legend.
- Figure 6A: x-axis legend need to be changed to the same as in Figure 5A.
- Figure 9A and 9B: drawing a dotted line crossing the 50% value in y-axis would help to visualize the explanation of the results from the text.
- Line 440: 8, 16, and 64 is unnecessary information since data from these time points are not shown.

Author Response
Dear reviewers:
Re: Manuscript ID: Microorganisms-1442003 and Title: CsrA regulates swarming motility, carbohydrate and amino acid metabolism in Vibrio alginolyticus.
Thank you for your comments concerning our manuscript entitled “CsrA regulates swarming motility, carbohydrate and amino acid metabolism in Vibrio alginolyticus” (Microorganisms-1442003). Those comments are valuable and very helpful. We have read it through carefully and made the revision as you suggested. The revised manuscript was resubmitted. The responses to the reviewer's comments were attached.
We highly appreciate your time and consideration.
Sincerely.
Bing LIU

Reviewer 2 Report
The manuscript by Lui et al.explores the role of CsrA in V.alginolyticus.
Overall, the manuscript is well written and clearly presents the data. Some of them, however, appear to be rather unnecessary and superfluous, in particular the sequence alignment and subsequent phylogenetic analyses of CsrA orthologs across several bacterial genera and species, many of which are not closely related to Vibrio. Thus, I'd encourage authors to remove or, at least, strongly reduce such parts, including Fig.1.
Data reported in Fig.4 are very interesting, but they would be strengthened by gene expression analyses, so as to get insights into the response to each carbon or nitrogen source. I think that the authors should include these data for a more comprehensive description.
Minor comments:
-L52: please correct into cholerae.
-Tab.3 and 4: given the size, due to the high number of primer pairs and genes showed, please consider moving them to supplementary data.
-Fig.3A: swarming
-L461: Vibrio, in Italics
Author Response

(The authors gave the same response as above.)

Round 2
Reviewer 2 Report
The manuscript has been revised according to the comments.
As for the gene expression analysis with different carbon sources, RNA could have been extracted taking into account the growth curves with each carbon source (i.e in the early or late exponential phase)
Author Response
Dear reviewers:
Re: Manuscript ID: Microorganisms-1442003 and Title: CsrA regulates swarming motility, carbohydrate and amino acid metabolism in Vibrio alginolyticus.
Thank you for your letter and the reviewers’ comments concerning our manuscript. Based on your comment and request, we have made clarification below. The responses to the reviewer's comments are marked in red and presented following. We highly appreciate your time and consideration.Please see the attachment.
Sincerely.
Chan Chen

This manuscript is a resubmission of an earlier submission. The following is a list of the peer review reports and author responses from that submission.